# Effect of future forest management on carbon and energy budgets in pine forests on mineral soil in southern Finland

Juha Leskinen<sup>1</sup>, Leif Backman<sup>1</sup>, Tiina Markkanen<sup>1</sup>, Jussi Lintunen<sup>2</sup>, and Tuula Aalto<sup>1</sup>

<sup>1</sup>Finnish Meteorological Institute, Helsinki, Finland

**Correspondence:** Juha Leskinen (juha.leskinen@fmi.fi)

Abstract. In forest management, carbon sequestration is often the main topic of interest when it comes to climate change mitigation. Here, we examined other climate impacts of forests - caused by changes in albedo, latent heat flux and sensible heat flux – and assessed their significance in relation to carbon sequestration in Finnish pine forests growing on mineral soil. Three different forest management scenarios with different harvest intensities were modelled with JSBACH\_FOM forced with data from three climate models CanESM2, MIROC5 and CNRM CM5 under two climate scenarios RCP 4.5 and RCP 8.5 from 2010 to 2054. Forest management scenarios were compared with each other, and the differences in the 45-year mean energy balance, connected to the relevant climate impacts, were converted from W m<sup>-2</sup> to carbon equivalent with time-dependentemission-equivalent (TDEE) metric. Based on our findings, carbon sequestration acts as the main driver behind the differences between the harvest scenarios in their total climate impact. Carbon sequestration was highest in the scenario with the lowest harvest intensity, and lowest in the scenario with the highest harvest intensity. The total carbon pool of the forest experienced growth in all of the harvest scenarios. The lowest harvest intensity yielded the highest carbon sequestration, but resulted in the highest absorption of solar radiation due to the low albedo, which slightly decreased the benefit from the carbon sequestration. The albedo-induced carbon equivalent climate impact differences between the harvest scenarios were in the range of 4-7% of the differences in carbon sequestration. The carbon equivalent differences from fluxes of latent and sensible heat between the harvest scenarios were in the range of 3-5% in comparison to the differences in carbon sequestration, with an opposite sign in relation to the differences from the albedo-induced climate impact. Accounting for the carbon equivalent climate impacts did not result in any of the scenarios becoming a carbon source, indicating that the forests remained beneficial in terms of climate change mitigation despite the management actions analysed in this study. The harvest scenario with lowest harvest intensity resulted in the most beneficial carbon equivalent climate impact in terms of climate change mitigation. After accounting for the differences in albedo and fluxes of latent and sensible heat in the total carbon equivalent climate impact between the scenarios, the climate change mitigation benefit resulting from the harvest scenarios with higher carbon sequestration was reduced in total by less than 5% across all harvest and climate scenarios. No significant differences were found between RCP 4.5 and RCP 8.5 in the climate impact differences from the different sources between the harvest scenarios.

<sup>&</sup>lt;sup>2</sup>Natural Resources Institute Finland, Helsinki, Finland

#### 25 1 Introduction

50

It is well established that forests have a potential to cool the Earth's climate by capturing and storing carbon dioxide from the atmosphere, which is the most important greenhouse gas making up for 53% of the climate warming caused by human related activities since 1750 (Minx et al., 2021). However, forests impact the Earth's climate and local surface energy budget in other ways as well. Betts (2000) showed that high absorption of solar radiation by forests due to their low albedo could offset the cooling benefit from carbon sequestration. Multiple studies have since confirmed (e.g. Bonan (2008); Lutz and Howarth (2014); Hovi et al. (2016); Bright et al. (2017); Luyssaert et al. (2018); Kellomäki et al. (2021)) that albedo should not be ignored when estimating the climate impacts of forest management actions. Forests have an important role in the water cycle as well (Bonan, 2008). The evapotranspiration rate of forests and the resulting evaporative cooling via latent heat flux is higher in comparison to other land cover types, as long as the soil water availability supports the high evapotranspiration rate (e.g. Madani et al. (2017); Zhang et al. (2023)). The relative significance of these processes varies across latitudes, climates and vegetation types, as the period of snow cover and length growing season can vary significantly in different geographic areas. It is thus important to research the significance of these climate impacts locally.

Forests and especially coniferous forests have a relatively low albedo when compared to other land cover types. Albedo values for Finnish pine, spruce and birch forests are 0.11, 0.08 and 0.17, respectively (Kuusinen et al., 2016). Pine and spruce albedo values are lower than those of for example grass (0.19–0.26) or sand (0.18–0.37) (Kondratyev, 1969). Total albedo of a forest is determined by tree species, forest structure, ground vegetation type and soil type (Kuusinen et al., 2014). Snow cover has a significant impact on winter albedo (Ni and Woodcock, 2000; Kuusinen et al., 2012). Albedo value for flat snow cover ranges from 0.5, for old and grainy snow, to 0.9 for fresh and fine snow (Wiscombe and Warren, 1980). Snow does not cover forest canopies perfectly as snow accumulates unevenly in comparison to other land cover types. Resulting from this, snow covered forests have lower winter albedo in comparison to snow covered non–forested areas (Kondratyev, 1969). The strength of the snow-albedo feedback is expected to decrease in the future, as the snow cover decreases with the warming climate (Pitman et al., 2011), which will potentially reduce the differences in winter albedo between forests and other land cover types. When trees are removed from the forest, more of the forest floor is exposed to direct sunlight. This increases the mean albedo of the area, leading to a local cooling effect (Lukeš et al., 2013; Kuusinen et al., 2014).

The impact of harvest on the water budget of a forest is not straightforward. Removal of trees reduces the leaf area index (LAI) and thus the surface area of transpiration, but the decrease in area can be partly compensated by the transpiration rate of the remaining trees increasing, as competition for water is reduced (e.g. Park et al. (2017); Wang et al. (2020); Zhang et al. (2023)). However, if pre-harvest evapotranspiration is not limited by water availability, evapotranspiration and the resulting evaporative cooling of the entire forest is expected to decrease after harvests (Wang et al., 2020). Removal of trees also removes shade from the remaining trees, which can elevate the rate of photosynthesis and the water use efficiency of the remaining trees (Gebauer et al., 2014).

This modelling study focuses on how different forest management practises can influence the climate change mitigation potential of Finnish pine forests growing on mineral soil. Pine forests growing on mineral soil were selected because pine is

the most common tree species in Finland (50% of the total wood volume) and two thirds of the forests in Finland are growing on mineral soils (Vaahtera et al., 2023). Finnish pine forests are representative of the boreal forest biome, which covers forests between approximately 50°N and 70°N characterized by coniferous trees such as pine, spruce and larch (Kayes and Mallik, 2020). Boreal forests are important for the global carbon sink as they make up 22% of the total forest carbon sequestration (Pan et al., 2011) and cover 27% of the global forest area (Kayes and Mallik, 2020).

We compare the relative significance of the most important climate impacts that are influenced by forest management in terms of their carbon equivalent values. Simulations of three forest management scenarios were carried out over a 45-year time period 2010–2054, driven with data from three climate models forced with two climate scenarios. Two climate scenarios were used in order to determine whether the relative significance of the climate impacts of the different forest management strategies will vary based on the trajectory of the climate change.

#### 2 Methods

## 70 2.1 Model description

The simulations were carried out with a forest management (FOM) version of the JSBACH land ecosystem model (JSBACH\_FOM) (Tyystjärvi et al., 2024). JSBACH is a process based land surface model that was originally a land component in MPI ESMs (Reick et al., 2020). The model can conserve energy, water and carbon balances of the system to correctly represent the real world. JSBACH\_FOM can be setup to represent different habitats by altering the model parameters related to soil and vegetation properties. The upper limit for the seasonally varying LAI is set dynamically based on the available leaf biomass and plant functional type (PFT), which makes it possible to simulate forest regrowth and ageing. The user can define forest age and rotation lengths to simulate different harvest scenarios and harvesting is done through clear-cuts. Meteorological driver data is required as input for the model, as we use it in stand-alone mode without an atmospheric circulation model. The model was run with daily inputs and outputs and 30-minute internal time-step.

#### 2.1.1 Water budget modelling

80

Water budget calculations of JSBACH\_FOM are separated in two parts, above and below ground water. Above ground water consists of snow and water on the canopy and at the surface. Snow cover is determined by the amounts of snowfall, sublimation, melting and wind-blow for both canopy and ground snow. The amount of liquid water is initially based on rainfall, which increases the above ground water budget. Rainfall is either absorbed by the soil or lost due to surface run-off and drainage. Water is constantly being removed from the water budget as the result of evaporation and transpiration, magnitude of which depends on the current rate of photosynthesis. The maximum amount of water that the soil can hold is given in JSBACH\_FOM by the field capacity. If the soil water content would exceed the field capacity, surplus will be moved to drainage flux, where it exits the grid cell through the bottom of the simulated soil column and is removed from the water budget.

#### 2.1.2 Albedo and radiation balance

O Photosynthetically active solar radiation (PAR) absorbed by the canopy is determined based on soil and vegetation type and LAI

The albedo for light in the visible (VIS) and near infrared (NIR) ranges are calculated based on snow cover, LAI and vegetation distribution. VIS and NIR albedo values in JSBACH\_FOM are determined based on PFT and soil type. Albedo is considered separately for snow covered surfaces, bare soil and vegetation cover. Surface types are divided into four categories; land not covered by vegetation canopy or snow, land covered by snow but not by vegetation canopy, vegetation canopy not covered by snow, and vegetation canopy covered by snow. Snow albedo varies based on the age of snow, temperature and if it covers soil or vegetation.

#### 2.1.3 Modelling of carbon balance

Gross primary productivity (GPP) available for vegetation functions and growth is based on the strength of photosynthesis, which depends on PAR,  $CO_2$  concentration and temperature, obtained from driver data. First, JSBACH\_FOM calculates the potential GPP, which represents the GPP in the absence of water stress. This is recomputed into actual GPP with hydrology module, accounting for the water stress. A fraction of GPP is used for vegetation maintenance respiration  $R_m$  and growth respiration  $R_g$ . Deducting  $R_m$  and  $R_g$  from GPP yields net primary productivity (NPP), describing the carbon accumulated in vegetation. Vegetation growth resulting from NPP is allocated to three carbon pools according to corresponding coefficients;  $C_W$ , consisting of woody parts, stems, branches.  $C_G$ , consisting of living parts, fine roots and vascular tissues.  $C_R$ , consisting of sugars and starches kept as energy reserve. The allocation routines also account for transfer of living biomass to litter carbon pools. JSBACH\_FOM has a maximum limit for carbon stored in  $C_W$ , while  $C_G$  and  $C_R$  are limited by the current LAI.  $C_G$  loses carbon due to grazing by herbivores and litter production via shedding, which is based on the seasonal cycle as well as vegetation type and age for  $C_G$ . For  $C_W$  and  $C_R$ , carbon loss resulting from shedding is modelled based on the turnover rate. For  $C_W$ , the turnover rate is on the order of decades while for  $C_R$  it is set at one year. Carbon is released from soil via heterotrophic soil respiration. Accounting for this yields the net ecosystem exchange (NEE), representing the total transfer of carbon between surface and the atmosphere. Litter and soil carbon in JSBACH\_FOM is handled by Yasso litter and soil decomposition model, version based on Yasso07 (Tuomi et al., 2009; Goll et al., 2015).

Yasso implementation in JSBACH\_FOM has five generic carbon pools that are divided based on their chemical quality, acid hydrolyzable, water soluble, ethanol soluble, neither hydrolyzable nor soluble (these four are also referred as AWEN pools) and humus. AWEN carbon pools are divided into above and below ground parts. These nine pools are further divided in two, based on if the carbon originates from organic woody matter  $C_W$  or from non-woody organic matter either from  $C_G$  or  $C_R$ .

These 18 pools are calculated independently, as carbon in each pool decomposes at a different rate.

## 2.2 Model setup and calibration

The JSBACH\_FOM was initially configured to use the "extra-tropical evergreen trees" PFT. These settings were calibrated further in order to represent Finnish pine forests growing on mineral soil. This was achieved with the data from SMEAR II forest station located in Hyytiälä, Finland (61.85°N, 24.3°E; see Figure 1). The SMEAR II is located within the boreal forest biome with pine trees being the dominant tree species. Vegetation and soil properties in the SMEAR II station are well documented in conjunction with the history of forest management practises (Kolari et al., 2022).

**Figure 1.** Location of the Hyytiälä SMEAR II forest station (black dot) and the pine wood volume (Luke, 2021) as green gradient on the map of Finland.

#### 125 2.2.1 Climate data

130

Representative climate data is required, as it is the most important factor behind the land-atmosphere interactions. Climate variables used as driver data in JSBACH\_FOM are air temperature, CO<sub>2</sub> concentration, air humidity, wind speed, precipitation, incoming solar radiation and downwelling longwave radiation. For the model calibration phase, hourly ERA5-Land climate data was used (Hersbach et al., 2020; Muñoz Sabater et al., 2021). ERA5-data was acquired for coordinates 61.8°N, 24.3°E, which is the closest available data point to the SMEAR II station. For the harvest scenario simulations from 2010 to 2054, the model was driven by downscaled and bias-corrected meteorological data from the EURO-CORDEX initiative under two climate scenarios, RCP 4.5 and RCP 8.5 (Jacob et al., 2014). The closest grid point to Hyytiälä in the EUR-44 domain was used.

Downscaled and bias-corrected data from CanESM2 (Chylek et al., 2011), MIROC5 (Watanabe et al., 2010) and CNRM\_CM5 (Voldoire et al., 2013) were used. Before downscaling and bias-correction, data from these models fell in the average range in terms of projected temperature change under both RCP 4.5 and RCP 8.5 within the CMIP5 climate–model family (Ruosteenoja et al., 2016).

## 2.2.2 Soil properties

Soil properties in JSBACH\_FOM determine how water and heat are absorbed and transported in the soil, which has a direct impact on vegetation growth and the surface energy budget. Soil parameters (Supplementary Table S1) for the model runs are based on the SMEAR II site characteristics data by Kolari et al. (2022), as well as on the JSBACH\_FOM built-in parameters (provided by (Hagemann and Stacke, 2015)). Soil type was set to loam, based on the distribution of clay, sand and silt (Ditzler et al., 2017)).

#### 2.2.3 Albedo

The albedo in JSBACH\_FOM for different land surfaces (canopy, litter, soil and snow) is set separately for radiation in the visible and near infrared range. Albedo values for these surfaces are available as predetermined values in JSBACH\_FOM for extra-tropical evergreen trees and used (Table 1), except for the visible light albedo. It was changed from 0.04 to 0.03, based on work by Kuusinen et al. (2014) to better represent a Finnish pine forest.

**Table 1.** Albedo values used for different land cover types in JSBACH\_FOM. Snow albedo varies, as temperature and age of snow both affect the snow surface.

| Land cover type | Visible albedo | Near infrared albedo |  |  |  |
|-----------------|----------------|----------------------|--|--|--|
| Pine canopy     | 0.03           | 0.23                 |  |  |  |
| Litter          | 0.10           | 0.16                 |  |  |  |
| Soil            | 0.11           | 0.20                 |  |  |  |
| Snow            | 0.52 - 0.90    | 0.30 - 0.65          |  |  |  |

### 2.2.4 Carbon sequestration

150

The carbon sequestration rate of vegetation depends on incoming solar radiation, water availability and LAI. There are also parameters affecting carbon sequestration, which differ between plant species. Maximum carboxylation rate determines the maximum rate of photosynthesis per leaf area, and a value of  $62.5 \times 10^{-6}$  mol  $CO_2$  m<sup>-2</sup> s<sup>-1</sup> was used in accordance with information from Kattge et al. (2009). Electron transport rate at 25 Celsius was set to  $118.8 \times 10^{-6}$  mol m<sup>-2</sup> s<sup>-1</sup> for extratropical evergreen trees. This value is the maximum carboxylation rate multiplied by a coefficient of 1.9, a value used in the land cover library of JSBACH\_FOM to determine maximum electron transport rate based on the maximum carboxylation rate. Specific leaf area is the ratio between leaf biomass and leaf area, which was set to 0.148 m<sup>2</sup> leaf mol<sup>1</sup> C (Goude et al., 2019).

#### 2.2.5 Forest growth equations

Forest growth in JSBACH\_FOM is governed by two equations describing the relationship between tree biomass, leaf biomass and the number of trees. Numerical values for the constants in the equations were derived based on the work of Marklund (1988) with some adjustments during model calibration, because in the initial model runs the growth was slower than expected based on the historical SMEAR II-data. In equation 1, the carbon content of leaves or needles  $C_{leaf}$  is determined based on the biomass per tree  $BM_{ind}$ . Equation 2 sets the relationship between the total vegetation biomass  $ln(BM_{veg})$  and the number of trees per hectare N, during times when  $C_G$  carbon pool is at its maximum value.

$$ln(C_{leaf}) = -1.98 + 0.72 \cdot ln(BM_{ind}) \tag{1}$$

165 
$$ln(BM_{veq}) = 15 - 0.39 \cdot ln(N)$$
 (2)

As the biomass of individuals increases in JSBACH\_FOM, self thinning of trees will occur, ratio given by eq. 2. When the competition for resources such as light and water increases, the environment can no longer support the original number of trees. In the model runs, the initial (and the maximum) number of individuals was set to 3000 per hectare, based on approximation of 2500 per hectare in the late 1960s in Hyytiälä, after a fresh forest was planted in 1962 (Kolari et al., 2022).

#### 170 2.2.6 Model output verification

The JSBACH\_FOM output was verified by comparing the modelled outputs of GPP, NEE, evapotranspiration and water balance to their respective values from SMEAR II (SMEARII, 2024). The core study stand in Hyytiälä was established in 1962 after a clear cut (Kolari et al., 2022). The stand went through thinning in 2002, where mainly pines smaller than average were removed. As a result of the 2002 thinning, the LAI of the canopy decreased by approximately 20% (Vesala et al., 2005). During 2019–2020, approximately 70% of pines with a diameter of less than 21 cm were harvested, together with approximately 25% of pines with a diameter of 21 cm (Aslan et al., 2024). This was replicated in the model calibration by dividing the area to three tiles, producing a comparable forest structure from 2014 to 2022 with harvests taking place in 2020 both in Hyytiälä and JSBACH\_FOM (Table 2).

**Table 2.** Modelled area in the verification run. The grid box was divided into three tiles, with 20%, 20% and 60% share. Forest is planted in 1962 and harvests are done in 2002 and 2020.

| Tile number | Area | 1962    | 2002    | 2020    |
|-------------|------|---------|---------|---------|
| 1           | 20%  | Planted | Harvest | Harvest |
| 2           | 20%  | Planted | _       | Harvest |
| 3           | 60%  | Planted | _       | _       |

185

For each individual tile, pines were planted in 1962. In 2002, harvest and replanting was performed for tile [1], which represents the 20% reduction in LAI observed in Hyytiälä. Tile [1] was harvested again in 2020, and the harvested trees approximate the 70% removal of pines with a diameter of less than 21 cm in Hyytiälä. In 2020, 25% of trees not harvested since 1962 are located in tile [2] and 75% in tile [3]. Harvest of tile [2] in 2020 represents the 25% reduction in the number of pines with a diameter of more than 22 cm in Hyytiälä. Values of GPP and NEE are impacted similarly in JSBACH\_FOM and in-situ measurements by the 2020 harvests (Supplementary Figures S1a and S1b). Soil water content before the harvest is similar between the two and it decreases in both JSBACH\_FOM and Hyytiälä, but the decrease is more prominent in Hyytiälä (Supplementary Figure S1d). The evapotranspiration rate remains higher in the model both before and after the harvest, with neither the model nor observations having a clear signal after the harvests took place (Supplementary Figure S1c). The calibrated JSBACH\_FOM is used for years 2010–2054 to simulate the different harvest scenarios, which are introduced in section 2.4.

## 190 2.3 CO<sub>2</sub> equivalent forcing

A metric is needed to quantify differences in climate impacts of forests both between the harvest and climate scenarios, and among the individual climate impacts within each scenario. This is achieved by converting the difference in radiative forcing at the surface, resulting from differences between the harvest scenarios, from W  $m^{-2}$  to its carbon equivalent value. This enables direct comparison with the carbon sequestration itself in terms of their relative climate change mitigation potential. Time-dependent emission equivalence (**TDEE**), first presented by Bright et al. (2016) and further examined by Bright and Lund (2021) was selected as the metric (see Equation 3).

$$\mathbf{TDEE} = A_E^{-1} \cdot k_{CO_2}^{-1} \cdot \mathbf{Y}_{CO_2}^{-1} \cdot \mathbf{RF}_{\Delta\alpha}^* \tag{3}$$

**TDEE** is a column vector representing yearly  $CO_2$  equivalent emission pulses required to produce a specific radiative forcing profile over multiple years, resulting from any source, such as difference in albedo between two forest management scenarios. The sum of the yearly  $CO_2$  equivalent emissions ( $\sum$  **TDEE**) represents the total  $CO_2$  equivalent emissions over the modelled time period.  $A_E$  is the surface area of the Earth,  $k_{CO_2}$  (Equation 4) is the global mean radiative efficiency of  $CO_2$ .  $Y_{CO_2}$  is a lower triangular matrix, where column and row values control the remaining fraction of the yearly  $CO_2$  equivalent pulses emitted to the atmosphere.  $Y_{CO_2}$  accounts for atmospheric decay of  $CO_2$  and adjusts the upcoming  $CO_2$  equivalent pulses accordingly, in order to keep the total  $CO_2$  equivalent level from the  $CO_2$  equivalent pulses at the correct level, where the decay of previous pulses is compensated by the following pulses.

 $\mathbf{Y}_{CO_2}$  is constructed based on an impulse response function (Supplementary Equation S1 and Supplementary Table S2) by Joos et al. (2013), value of which at a given time represents the fraction of atmospheric  $CO_2$  pulse remaining in the atmosphere after time (t) in years.  $\mathbf{RF}_{\Delta\alpha}^*$  is a column vector, representing the yearly mean radiative forcing, which **TDEE** converts from W m<sup>-2</sup> to represent yearly  $CO_2$  equivalent emissions. In this study,  $\mathbf{RF}_{\Delta\alpha}^*$ , results either from the differences in surface albedo or in the fluxes of latent and sensible heat between the harvest scenarios. Instead of using yearly mean radiative forcing

values resulting from the differences between the harvest scenarios in  $\mathbf{RF}_{\Delta\alpha}^*$ , 45-year mean values are used, as the cumulative differences during 2010–2054 are the main focus of this study.

The global mean radiative efficiency is a function of four parameters:

$$k_{CO_2}^{-1} = \frac{\alpha_{CO_2} \cdot \epsilon_{air} \cdot 10^6}{\epsilon_{CO_2} \cdot M_{atm}} \tag{4}$$

where  $(\epsilon_{air})$  and  $(\epsilon_{CO_2})$  are molecular weights of air and  $CO_2$ , respectively.  $M_{atm}$  is mass of the atmosphere and  $\alpha_{CO_2}$  (Equation 5) is the radiative forcing resulting from if the atmospheric  $CO_2$  concentration were to increase by 1 ppm compared to its current value. The more  $CO_2$  there is in the atmosphere, the less sensitive the atmosphere is to additional increases of the  $CO_2$  concentration in terms of radiative forcing (Joos et al., 2013), which is an important distinction as we are running the model under both RCP 4.5 and RCP 8.5.

$$220 \quad \alpha_{CO_2} = 5.35Wm^2 \cdot \ln\left(\frac{(\text{ppm+1})}{\text{ppm}}\right) \tag{5}$$

In **TDEE**, the CO<sub>2</sub> equivalent pulses produce a radiative forcing level corresponding to the 45-year mean of  $\mathbf{RF}_{\Delta\alpha}^*$  immediately starting from the first year. Resulting from this, the first CO<sub>2</sub> equivalent pulse is higher than the following pulses. The following pulses mainly compensate for the decay rate of the previous pulses. Because the initial CO<sub>2</sub> equivalent pulse is strongest, atmospheric conditions present during the first simulation year of 2010 are heavily weighed in the total CO<sub>2</sub> equivalent climate impact during 2010–2054. In order to mitigate this, atmospheric CO<sub>2</sub> concentrations present during 2010–2054 under RCP 4.5 and RCP 8.5 were averaged and used from 2010 to the final simulation year of 2054. First, yearly values for  $\alpha_{CO_2}$  were calculated with the corresponding CO<sub>2</sub> concentration levels under both RCP 4.5 and RCP 8.5. Because of the logarithmic nature of  $\alpha_{CO_2}$ , it was important to calculate yearly values for  $\alpha_{CO_2}$  instead of using the mean CO<sub>2</sub> concentration level under RCP 4.5 and RCP 8.5 to determine the mean 2010–2054 value for  $\alpha_{CO_2}$ . The mean of the 45 yearly  $\alpha_{CO_2}$  values was then used to determine  $k_{CO_2}$  separately for RCP 4.5 and RCP 8.5, at 1.5595 ·10<sup>-15</sup> W m<sup>-2</sup> kg<sup>-1</sup> and 1.4952 ·10<sup>-15</sup> W m<sup>-2</sup> kg<sup>-1</sup>, respectively.

## 2.4 Harvest scenarios

The three harvest scenarios used are derived using the Finnish Forest and Energy Policy (FinFEP) model (Lintunen et al., 2015). The scenarios depict the harvest behaviour of three types of forest owners that follow ecologically oriented, balanced, and profit-oriented management strategies. The national harvest levels follow the policy scenario of the Finnish Climate and Energy Strategy of 2016 up to 2030, reaching 79 million m<sup>3</sup> and slowly increasing further after 2030. In the scenarios, Pines are grouped into five-year age classes, from 5-year to 155-year and older. The management actions in the simulations are performed with 5-year intervals, starting from 2010 with the final management action occurring in 2050 while the simulations are ran until the end of 2054. The fractions of land area in each age class in 2010 and 2050 are presented in the Figures 2, 3,

and 4 in the different harvest scenarios. More in-depth visualization of the fraction of trees in each age class during 2010–2050 can be found from the Supplementary Figure S2.

#### 2.4.1 Ecologically oriented management

In the ecologically oriented harvest scenario (EC), rotation lengths are longest among the harvest scenarios. Harvests are mainly targeted at old trees, as can be seen from the age-class distribution in Figure 2, where the age class of 155 years or older comprises more than 10% of the total. Approximately 20% of the land area is in the 35- and 40-year-old age classes in 2010. After 40 years in 2050, these 35- and 40-year age classes represent the 80- and 85-year-old age classes, which are the three largest age classes together with the trees older than 155 years. EC is used as a baseline when comparing differences in the climate change mitigation potential of the harvest scenarios, resulting from the different climate impact sources.

Figure 2. Forest age class distribution in 2010 (orange) and in 2050 (blue) in the ecologically oriented forest management scenario (EC).

## 2.4.2 Balanced management

In the balanced harvest scenario (BA), modest harvests are made for mature age classes (see Figure 3). Some pines are allowed to reach the age of more than 155 years, but the fraction is one-fourth of that in EC. Most trees are harvested at the age of 100 years. The fraction of young trees is higher in BA than in EC, as the harvested trees are replaced with new individuals. Similarly as in EC, the 35- and 40-year-old age classes have the highest fraction of trees in 2010, which also remain the highest fraction in 2050 when they are 80- and 85-year-old.

260

Figure 3. Forest age class distribution in 2010 (orange) and in 2050 (blue) in the balanced (BA) forest management scenario.

## 2.4.3 Profit-oriented management

In the profit-oriented management scenario (PR), harvests are conducted earlier than in the other harvest scenarios (see Figure 4). During the 40-year time period 2010–2050, the forest age distribution changes notably. The dominant age classes in 2050 are no longer the 35- and 40-year-old trees from 2010, as majority of the trees are harvested before reaching the age of 85 years. Instead, 10-year-old and younger trees make up for the largest age classes, followed by pines older than 55 years but younger than 80 years.

Figure 4. Forest age class distribution in 2010 (orange) and in 2050 (blue) in the profit oriented (PR) forest management scenario.

## 3 Results

The relative differences between the harvest scenarios in their different climate impact components are first examined in CanESM2 on a monthly mean level. The differences between the harvest scenarios in CNRM\_CM5 and MIROC5 are included later in the analysis of the 45-year differences in the different climate impact sources between the harvest scenarios.

## 3.1 Carbon sequestration

Modelled yearly mean of the total carbon pool (see Figure 5) experiences growth in all of the harvest and climate scenarios from 2010 to 2054, ranging from 0.2 kg C m<sup>-2</sup> in PR under RCP 8.5 to 3.1 kg C m<sup>-2</sup> in EC under RCP 4.5. BA falls between EC and PR under both RCP 4.5 and RCP 8.5. In all of the forest management scenarios the total carbon pool increase is higher under RCP 4.5 than under RCP 8.5. Initially, the total carbon pools decreases in all scenarios expect in EC under RCP 4.5. In both EC and BA, the total carbon pool sustains growth under both climate scenarios starting from 2013 and 2018, respectively. However in PR, the total carbon pool does not increase until 2022 and experiences a negative trend starting from 2045. The change in woody carbon pool and the sum of below and above ground AWEN carbon pools during 2010–2054 are displayed in Figure A1 and A2.

**Figure 5.** Change in the annual mean of total carbon pool from 2010 to 2054 in CanESM2 in ecological (EC, blue), balanced (BA, black) and profit oriented (PR, orange) harvest scenarios under RCP 4.5 (solid lines) and RCP 8.5 (dashed lines) climate scenarios.

Yearly average LAI (see Figure 6) is highest in EC throughout the modelled time period, followed by BA and PR under both RCP 4.5 and RCP 8.5. Fastest initial increase in LAI occurs in PR, where in 2030 it nearly reaches the level of LAI in BA, but the LAI in PR decreases after 2040. LAI in both EC and BA grows continuously during 2010–2054. In BA, LAI exhibits faster initial growth than in EC, but the growth rate levels to the rate seen in EC from 2035 onwards. Initially, the LAI values under RCP 4.5 are slightly lower than those under RCP 8.5, but the difference diminishes over the simulation period.

**Figure 6.** Annual mean values of leaf are index (one sided, per land area) during 2010–2054 in CanESM2 with EC (blue), BA (black) and PR (orange) under RCP 4.5 (solid lines) and RCP 8.5 (dashed lines).

## 3.2 Energy balance

# 280 3.2.1 Albedo

Monthly mean values of albedo averaged over 2010–2054 are highest during the winter months and lowest during the summer months in both visible (VIS) and near infrared (NIR) ranges, and the absolute values in EC under RCP 4.5 are displayed in Figure 7). Yearly maximum and minimum values of albedo occur at the same time of year in BA and PR as well, under both RCP 4.5 and RCP 8.5.

**Figure 7.** Monthly mean albedo values in EC during 2010–2054 under RCP 4.5 (solid lines) and RCP 8.5 (dashed lines), in CanESM2. Albedo in NIR (orange), VIS (blue) and VIS+NIR (black) ranges are displayed separately. VIS+NIR albedo is calculated based on the distribution of incoming VIS and NIR radiation (Figure B1).

Monthly mean albedo values in BA and PR are compared with EC in Figure 8 under RCP 4.5. Albedo values in both VIS and VIS+NIR are higher in both BA and PR in comparison to EC during the whole year, and the difference is most prominent during the winter months peaking in February, and lowest during the summer months. The difference between PR and EC is approximately two times larger than the difference between BA and EC throughout the year. NIR albedo values in BA and PR are higher than in EC only during January and February, while for the remainder of the year NIR albedo is highest in EC.

**Figure 8.** Monthly mean albedo values from 2010 to 2054 in CanESM2 of BA (dashed lines) and PR (solid lines) in comparison to EC under RCP 4.5, in CanESM2. Albedo differences in NIR (orange), VIS+NIR (black) and VIS (blue) are shown. Positive values indicate that albedo is higher in BA or PR than in EC in the respective month.

Monthly mean snow cover during 2010–2054 for land and canopy are displayed in figures 9 and 10. Land snow cover is highest in February, ranging from 38% in EC to 42% in PR under RCP 4.5, and from 30% (EC) to 35% (PR) under RCP 8.5. Canopy snow cover is significantly lower, being highest in January where it ranges from 4% to 6% depending on the scenario. There is none or very little snow cover from April to October, after which the monthly mean snow cover fraction increases until January for land or until February for canopy, before decreasing towards the summer. Snow cover is higher under RCP 4.5 than under RCP 8.5 in all months apart from December for both the land and canopy.

**Figure 9.** Monthly mean land now cover fraction during the simulation period 2010–2054 in the ecological (EC, blue), balanced (BA, black) and profit oriented (PR, orange) harvest scenarios under both the RCP 4.5 (solid lines) and RCP 8.5 (dashed lines) climate scenarios, in CanESM2

**Figure 10.** Monthly mean canopy now cover fraction during the simulation period 2010–2054 in the ecological (EC, blue), balanced (BA, black) and profit oriented (PR, orange) harvest scenarios under both the RCP 4.5 (solid lines) and RCP 8.5 (dashed lines) climate scenarios, in CanESM2.

The monthly mean albedo-induced impact on the radiative forcing between the harvest scenarios is displayed in Figure 11. EC absorbs more radiation than both BA and PR during every month of the year. The albedo-induced impact is least prominent from October to January, where the differences between the harvest scenarios are less than 0.1 W m<sup>-2</sup> between all of the scenarios. After January, the albedo-induced difference increases and the first maximum occurs in March, where EC absorbs approximately 0.2 W m<sup>-2</sup> and 0.4 W m<sup>-2</sup> more radiation in comparison to BA and PR, respectively. In April, the differences momentarily decrease, and from May to July the differences are comparable to those of May, peaking in June. The difference between RCP 4.5 and RCP 8.5 is most prominent early in the year, and under RCP 4.5 the differences between the harvest scenarios are mostly larger than under RCP 8.5. An exception is seen in EC-PR, where in March and April the differences are larger under RCP 8.5. Corresponding values under RCP 8.5 are 0.1236 W m<sup>-2</sup> and 0.2433 W m<sup>-2</sup>.

**Figure 11.** The albedo-induced monthly mean radiative forcing differences between the harvest scenarios during 2010–2054 in CanESM2. EC-BA (blue) and EC-PR (orange) are displayed for both under RCP 4.5 (solid lines) and under RCP 8.5 (dashed lines). Positive values indicate EC absorbing (reflecting) more (less) radiation than BA or PR.

#### 3.2.2 Latent heat flux

305

Monthly mean latent heat flux differences during 2010–2054 and the resulting variation in the evaporative cooling between the harvest scenarios is most prominent during the summer months, when EC has the highest latent heat flux. Differences are less prominent during the winter months, when latent heat flux is highest in PR (see Figure 12. Differences in relation to EC are more prominent in PR than in BA, with the monthly mean differences being approximately twice as large. There is no significant distinction between RCP 4.5 and RCP 8.5, expect for July and August, where the latent heat flux difference between EC and the other harvest scenarios is more than 10% higher under RCP 8.5 than under RCP 4.5. The 45-year mean difference in the latent heat flux under RCP 4.5 in comparison to EC is 0.819 W m<sup>-2</sup> and 1.402 W m<sup>-2</sup> in BA and PR, respectively, and 0.914 W m<sup>-2</sup> and 1.574 W m<sup>-2</sup> under RCP 8.5.

**Figure 12.** Differences in latent heat flux monthly means during 2010–2054 of BA (blue) and PR (orange) compared to EC, both under RCP 4.5 (solid lines) and RCP 8.5 (dashed lines) in CanESM2. Positive values indicate BA or PR having higher sensible heat flux directed towards the soil than EC, resulting in more heat being transferred away from the atmosphere.

#### 3.2.3 Sensible heat flux

The differences in monthly mean sensible heat flux during 2010–2054 are shown in Figure 13. From April to September, BA and PR experience more cooling via sensible heat flux in comparison to EC. The opposite is true from October to March, but the differences between the harvest scenarios are largest in magnitude during the summer months. As a result, when averaged across the whole year, EC experiences least cooling via sensible heat flux. The differences between PR and EC are approximately twice as large as between BA and EC. There are no large differences between the climate scenarios expect for in July and August, similarly to those in the latent heat flux. On average throughout the 45-year simulation period, EC experiences 0.714 W m<sup>-2</sup> and 1.207 W m<sup>-2</sup> less cooling compared to BA and PR, respectively, under RCP 4.5 due to the lower sensible heat flux. Under RCP 8.5, EC removes less heat from the atmosphere by 0.817 W m<sup>-2</sup> and 1.387 W m<sup>-2</sup> in comparison to BA and PR, respectively.

**Figure 13.** Differences in sensible heat flux monthly means during 2010–2054 of BA (blue) and PR (orange) compared to EC, both under RCP 4.5 (solid lines) and RCP 8.5 (dashed lines) in CanESM2. Positive values indicate BA or PR having higher sensible heat flux directed towards the soil than EC, resulting in more heat being transferred away from the atmosphere.

## 3.2.4 Combined energy balance components

Latent and sensible heat fluxes are closely linked to each other and are examined together in Figure 14. The differences in the combined flux identify EC as the climate scenario with the highest cooling effect, as both BA and PR experience less cooling than EC from February to September. The opposite occurs in November and December, but the magnitude of the differences during these months is insignificant in comparison to what is modelled between February and September. January and October have both positive and negative values for BA and PR when compared with EC, depending on the climate scenario.

**Figure 14.** Combined latent and sensible heat flux monthly mean differences during 2010–2054 in BA (blue) and PR (orange) compared to EC, both under RCP 4.5 (solid lines) and RCP 8.5 (dashed lines) in CanESM2. Positive values indicate BA or PR having higher combined heat flux than EC directed towards the soil, resulting in more heat being transferred away from the atmosphere.

The monthly mean differences between the harvest scenarios in the combined differences for albedo and fluxes of latent and sensible heat are displayed in Figure 15. The highest observed monthly mean difference is modelled for March, where EC experiences more cooling by approximately 0.2 W m<sup>-2</sup> and 0.4 W m<sup>-2</sup> in comparison to BA and PR, respectively. However, it is the only month where the combined surface energy balance is prominently on the side of EC experiencing more cooling. EC is warmest among the harvest scenarios from June to January. Under RCP 4.5, on average EC cools the atmosphere less by 0.105 W m<sup>-2</sup> and 0.195 W m<sup>-2</sup> in comparison to BA and PR in 2010–2054, respectively. Under RCP 8.5, these values are 0.097 W m<sup>-2</sup> and 0.187 W m<sup>-2</sup> for BA and PR, respectively.

**Figure 15.** Combined albedo-induced, latent and sensible heat flux monthly mean differences in radiative forcing during 2010–2054 in BA (blue) and PR (orange) compared to EC, under RCP 4.5 (solid lines) and RCP 8.5 (dashed lines) in CanESM2. Positive values indicate BA or PR having a higher heat flux than EC directed towards the soil, resulting in more heat being transferred away from the atmosphere and absorbed by soil.

#### 3.3 Water balance

The monthly mean volumetric soil moisture during 2010–2054 is highest in EC and lowest in PR from January to April, and the opposite is true from May to December, while the soil moisture in BA remains between EC and PR across the year (see Figure 16). The soil water content is higher under RCP 4.5 than under RCP 8.5 from February to May, and lower from June to January. The differences between both the harvest and climate scenarios are most pronounced during the summer months, and the differences between RCP 4.5 and RCP 8.5 are especially high in July, with the variation between the climate scenarios being more than three times higher than during any other month.

**Figure 16.** Monthly mean value of volumetric soil moisture during 2010–2054 in CanESM2 in EC (blue), BA (black) and PR (orange), under RCP 4.5 (solid lines) and RCP 8.5 (dashed lines).

## 3.4 Relative significance

355

The mean radiative forcing differences between the harvest scenarios, derived from the different climate impact sources and averaged over the simulation period 2010–2054, were converted into their respective carbon-equivalent values with the TDEE metric, representing the equivalent climate impact over the same time period. The carbon equivalent values SUM\_TDEE are displayed visually in Figure 17, where the bar height indicates how much higher (or lower, if negative) the carbon sequestration or carbon equivalent climate impact is in EC compared to BA or PR. Carbon equivalent total differences during 2010–2054 are also displayed numerically in Table 3, and corresponding radiative forcing values in Supplementary Table S3. For reference, yearly carbon equivalent profiles used to derive ∑TDEE for EC−BA under RCP 4.5 with CanESM2 are shown in Supplementary Figure S3.

Carbon sequestration from the atmosphere during 2010–2054 is higher in EC than in both BA and PR under both climate scenarios, and in all climate models.

In terms of albedo-induced radiative forcing, EC has disadvantage in terms of climate change mitigation when compared to BA and PR, indicating that it absorbs (releases) less (more) carbon equivalent from the atmosphere (ground) than the other harvest scenarios. The difference between EC and PR is higher than that of EC and BA, resulting in PR having the climate change mitigation advantage from the albedo component.

In terms of latent heat flux, EC has carbon equivalent climate change mitigation advantage in comparison to BA and PR.

In carbon equivalence, this advantage is approximately half as large as the advantage in carbon sequestration in EC, when EC is compared with BA, and approximately one quarter when EC and PR are compared. The carbon equivalent latent heat flux differences are slightly higher under RCP 8.5 than under RCP 4.5.

365

The differences in sensible heat flux between the harvest scenarios are of opposite sign and 10–20% lower in their absolute values than the differences in latent heat flux, varying based on the model and the climate scenario. Similarly to the differences in latent heat flux, the sensible heat flux differences are higher when EC is compared with PR than with BA, resulting in PR having the most beneficial effect in terms of climate change mitigation from the sensible heat flux differences. The carbon equivalent sensible heat flux differences between the harvest scenarios are higher under RCP 8.5 than under RCP 4.5, similarly to the differences in latent heat flux.

When the latent and sensible heat flux differences are combined, EC emerges as the harvest scenario with the highest (lowest) carbon equivalent removal (release) from the atmosphere (ground) in relation to the other harvest scenarios from this climate impact.

Combining the differences between the harvest scenarios in their carbon sequestration with their carbon equivalent climate impact differences from albedo and fluxes of latent and sensible heat, results in EC having the highest carbon equivalent removal from the atmosphere, followed by BA and PR.

**Figure 17.** Carbon equivalent climate impact differences during 2010–2054 in EC compared to BA (left column group) and PR (right column group) from the different sources; carbon sequestration (blue), albedo (orange), latent heat flux (green), sensible heat flux (red) and the sum of each source (purple). Bar height represents the 3-model-mean, and the uncertainty range displays the highest and lowest differences modelled by the three climate models; CanESM2, CNRM\_CM5 and MIROC5. Differences under RCP 4.5 are represented by the clear-colour bars and by the dashed bars under RCP 8.5. Positive values indicate that the carbon sequestration or carbon equivalent value is higher in EC, with carbon equivalent values representing EC absorbing (releasing) more (less) carbon from the atmosphere (ground).

**Table 3.** Carbon equivalent total climate impact in EC compared to BA and PR during 2010–2054 under both RCP 4.5 and RCP 8.5, in kg C m<sup>-2</sup>. Positive values indicate EC removing more carbon or carbon equivalent from the atmosphere in relation to BA or PR. Results from all three climate models (CanESM2, CNRM\_CM5 and MIROC5) are displayed separately, and their mean values in the last two rows.

|            | Carbon sequestration |         | Albedo  |         | Latent heat |         | Sensible heat |         | Total carbon |         |
|------------|----------------------|---------|---------|---------|-------------|---------|---------------|---------|--------------|---------|
|            | RCP 4.5              | RCP 8.5 | RCP 4.5 | RCP 8.5 | RCP 4.5     | RCP 8.5 | RCP 4.5       | RCP 8.5 | RCP 4.5      | RCP 8.5 |
| CanESM2    |                      |         |         |         |             |         |               |         |              |         |
| EC – BA    | 1.08                 | 1.08    | -0.0811 | -0.0820 | 0.519       | 0.604   | -0.452        | -0.540  | 1.07         | 1.06    |
| EC – PR    | 2.65                 | 2.64    | -0.155  | -0.161  | 0.887       | 1.03    | -0.767        | -0.919  | 2.62         | 2.59    |
| CNRM_CM5   |                      |         |         |         |             |         |               |         |              |         |
| EC – BA    | 1.06                 | 1.02    | -0.0672 | -0.0701 | 0.423       | 0.374   | -0.380        | -0.328  | 1.04         | 0.0996  |
| EC – PR    | 2.65                 | 2.61    | -0.131  | -0.135  | 0.691       | 0.488   | -0.608        | -0.402  | 2.60         | 2.56    |
| MIROC5     |                      |         |         |         |             |         |               |         |              |         |
| EC – BA    | 1.02                 | 1.06    | -0.0646 | -0.0674 | 0.404       | 0.432   | -0.367        | -0.398  | 0.992        | 1.03    |
| EC – PR    | 2.54                 | 2.64    | -0.126  | -0.134  | 0.563       | 0.674   | -0.494        | -0.607  | 2.48         | 2.47    |
| Model mean |                      |         |         |         |             |         |               |         |              |         |
| EC – BA    | 1.05                 | 1.05    | -0.0710 | -0.0732 | 0.449       | 0.470   | -0.400        | -0.422  | 1.03         | 1.02    |
| EC – PR    | 2.61                 | 2.63    | -0.138  | -0.143  | 0.714       | 0.731   | -0.623        | -0.643  | 2.56         | 2.57    |

#### 375 4 Discussion

380

The highest growth in total carbon pool, and thus in carbon sequestration occurs in EC. This is expected, as trees grow older in EC in comparison to the other harvest scenarios. While the total carbon pool in EC grows throughout the simulation, the total carbon pools in both BA and PR initially see a decline. This decline is explained by the timing of harvests and by the AWEN carbon pool, which decreases in all of the scenarios after the first few years, later stabilizing around 2030 before increasing (see Figure A2). From 2022 onwards, the total carbon pool in BA grows for the remainder of the simulation. The total carbon pool in PR has an upward trend from 2018, but it turns downwards after 2045. This occurs simultaneously with large portion of trees reaching the harvest age of 80 years in PR (see Figure 4), explaining the decrease in the total carbon pool.

Initial differences in 2010 between the harvest scenarios in their respective total carbon pools are explained by their LAI values. In EC, LAI is initially higher by approximately 20% and 50% in relation to BA and PR, respectively. In 2010, The age-class distribution of pines extends up to 155 years and older in EC, while trees older than 100 and 80 are scarce in BA and PR, respectively. The differences in LAI between the harvest scenarios are amplified by older pines having higher biomass per individual.

The highest monthly mean albedo values are modelled to occur during winter months and lowest during summer months in EC. This is in line with the expectation of snow cover resulting in higher winter albedo (Ni and Woodcock, 2000; Kuusinen et al., 2012). The modelled minimum and maximum monthly mean albedo values are lower than expected based on literature

420

425

(Kuusinen et al., 2012; Peräkylä et al., 2025). This can be attributed to the relatively low land and canopy snow cover during 2010–2054. Land snow cover values are at their highest around 40% in February under RCP 4.5, and 32% under RCP 8.5, varying based on the harvest scenario. For canopy snow cover, the maximum monthly mean value occurs in February and ranges from 4% to 6%. Thus, the fraction of modelled land and canopy area not covered by snow is relatively high even during the months of highest snow cover, which results in relatively low winter albedo. Snow cover is the only variable where the differences between the climate scenarios surpass the differences between harvest scenarios.

The monthly mean VIS and VIS+NIR albedo values during 2010–2054 were highest in PR and lowest in EC throughout the year. NIR albedo values are highest in EC from March to December and lowest in January and February. Seasonal variation and the absolute values in the NIR albedo differences are lower than the differences in VIS albedo between the harvest scenarios. Thus, differences in VIS albedo between the harvest scenarios is the main driver behind the differences in monthly mean VIS+NIR albedo. The differences in albedo between the harvest scenarios are most pronounced during the winter months, coinciding with the periods of highest snow cover. Snow cover amplifies the existing albedo differences resulting from the differences in land cover between the harvest scenarios. In the absence of snow cover, albedo differences between the harvest scenarios remain largely unchanged from April to October. LAI, and thus the poorly snow covered canopy area, is higher in EC than in both BA and PR, resulting in lowest snow-albedo among the harvest scenarios.

In order to examine the seasonal variability in the monthly mean VIS albedo values during 2010-2054, an experiment was performed where a fully grown forest undergoes a clear-cut (see Figure C1). In the harvest scenarios, only a fraction of each age class is harvested, leading to relatively small changes in albedo. In this experiment, annual mean LAI values of 2.6 m<sup>2</sup>  $\mathrm{m}^{-2}$  were present for 10 years during 2000–2009 prior to the clear-cut, followed by a 10-year period with LAI below 0.1  $\mathrm{m}^2$ m<sup>-2</sup> in 2010–2019, representing non-forested area. Before the clear-cut, 10-year monthly mean VIS albedo values were lower by approximately 0.01 than those in EC across the year. After the clear-cut in this experiment, the highest monthly mean VIS albedo value rose significantly and occurs in February at 0.21, which is an increase of 0.14 in comparison to the pre-clear-cut value of 0.07. The highest monthly mean VIS albedo value in EC during 2010-2054 is 0.09 (see Figure 7 and in PR it is higher than that of EC by 0.015 (see Figure 8 due to less canopy cover. In PR, the mean LAI during 2010–2054 is approximately 1.4 m<sup>2</sup> m<sup>-2</sup>, and 1.8 m<sup>2</sup> m<sup>-2</sup> in EC. Even though these LAI values are substantially lower than the 2.6 m<sup>2</sup> m<sup>-2</sup> of the fully grown forest before the clear-cut, the highest difference in VIS albedo between PR and the fully grown forest of the experiment is only 0.035 (0.07 vs. 0.105), occurring in February. This suggests that even a moderate canopy cover – such as that in PR with LAI of 1.4 m<sup>2</sup> m<sup>-2</sup> – is enough to reduce VIS albedo during winter by a significant margin in comparison to a non-forested area. While LAI in PR during 2010-2054 is approximately 55% of that in the fully grown forest, the VIS albedo value in PR does not fall between the VIS albedo values of a fully grown forest and a non-forested area, it is much closer to the fully grown forest. The additional experiment covered different time periods (2000-2009 & 2010-2019) than the main simulations of this study (2010–2054) so the results are only indicative, but provide useful insight on the albedo values modelled in this study.

Even though the monthly mean differences in albedo between the harvest scenarios are highest in February, the monthly mean differences in absorbed radiation are at their highest in March and from May to July. This stems from the annual pattern of incoming radiation, which has significant annual variation (Figure B1). While the incoming radiation has its maximum

455

460

value in June and is twice as high as in March, the differences in albedo between the harvest scenarios are simultaneously two times lower in June in comparison to March. In February, the differences in albedo between the harvest scenarios are at their maximum, but the incoming radiation is near its minimum value. Therefore, the low incoming radiation is insufficient to drive a more significant albedo—induced difference in the radiative forcing between the harvest scenarios. This finding puts emphasis on the importance of latitude, as the ratio of incoming solar radiation between the summer and winter months determines how important the role of winter albedo is in terms of its impact on the annual surface energy balance. The local climate is also important in conjunction with the latitude, since the amount of precipitation and typical duration and timing of snow cover can vary greatly even among the same latitude.

The differences in monthly mean volumetric soil moisture between the harvest scenarios are in line with the assumption that forests are areas of high evapotranspiration (Madani et al., 2017). In EC, volumetric soil moisture is lowest and it also decreases with the fastest rate in spring. There is a relatively large difference in volumetric soil moisture between RCP 4.5 and RCP 8.5 in July. Under RCP 8.5, water availability from precipitation is higher (Ruosteenoja et al., 2020), and evapotranspiration is limited less by soil moisture content than under RCP 4.5.

The highest latent heat flux during 2010–2054 occurs in EC. Latent heat flux in PR is highest from October to March, but the difference between EC and PR is less significant in comparison to what it is from April to September, when EC has the highest latent heat flux. During the winter months, photosynthesis ceases and transpiration from the canopy decreases (Supplementary Figure S4). However, evaporation from the land surface continues and is the main driver behind evapotranspiration during winter. PR has the lowest LAI and canopy cover, thus having the highest bare soil evaporation (Supplementary Figure S5). The differences in latent heat flux between the harvest scenarios are relatively similar under both RCP 4.5 and RCP 8.5 during all months apart from July and August, where the latent heat flux differences under RCP 8.5 are more pronounced. This occurs simultaneously with the highest soil moisture differences between RCP 4.5 and RCP 8.5, indicating that evapotranspiration and the resulting latent heat flux are limited to some extent by water availability during the summer months.

While not identical, the differences in the monthly mean values of sensible heat flux between the harvest scenarios during 2010–2054 follow a similar annual pattern with the differences in latent heat flux. This occurs because evapotranspiration reduces the surface temperature via evaporative cooling, which impacts the sensible heat flux by altering the temperature gradient between surface and the atmosphere.

When fluxes of latent and sensible heat are combined, EC has a net cooling effect from the two combined fluxes, which remains at a constant level from February to July. The only exception is March, where the difference is two times higher than in any other month, which in JSBACH\_FOM is related to snow melting. The differences in latent heat flux between the harvest scenarios are not entirely counterweighted by the differences in sensible heat flux, and the remaining margin is approximately half of the carbon equivalent climate impact from the albedo–induced radiative forcing differences. This counterweights the albedo–induced differences in the climate change mitigation potential between the harvest scenarios.

EC has the highest total carbon equivalent climate change mitigation benefit from the studied climate impacts; carbon sequestration and carbon equivalent impacts from albedo, latent heat flux and sensible heat flux. Carbon sequestration is the main driver behind the total climate impact differences between the harvest scenarios. While BA and PR do provide a

carbon equivalent benefit with their higher albedo, in comparison to EC, it only offsets the climate impact of higher carbon sequestration in EC by between 5% and 7%. Individually, the differences in fluxes of latent and sensible heat between the harvest scenarios approach half of the climate impact of carbon sequestration when EC and BA are compared, and one fourth with EC and PR. However, since the differences in fluxes of latent and sensible heat act to counterweigh each other, the differences between the harvest scenarios in the combined flux ranges from 3% to 5% in comparison carbon sequestration.

The climate change mitigation benefit of EC in relation to BA and PR is slightly mitigated when the other climate impacts are evaluated together with carbon sequestration. This reduction is most pronounced when BA is compared with EC under RCP 4.5, where the total climate impact benefit of EC in comparison to BA is reduced to 96% of what it would be if only carbon sequestration was accounted for. Reduction is least significant when PR is compared with EC under RCP 4.5, where the climate change mitigation benefit of EC from only accounting for carbon sequestration is reduced to 98% by accounting for the other climate impacts as well.

The differences between the harvest scenarios in their climate impacts are relatively similar under both RCP 4.5 and RCP 8.5. The differences in the results between the climate scenarios are smaller than the model-to-model variation between the data from CanESM2, CNRM\_CM5 and MIROC5, and the model data themselves were in the average range in the CMIP5 climate—model family. Most pronounced differences between RCP 4.5 and RCP 8.5 are in latent heat flux and soil moisture during summer and in snow cover. Differences in LAI and total carbon pool remain small between the climate scenarios. The conditions under the climate scenarios are relatively close to each other initially in 2010, diverging further with time. Higher differences could arise if the simulation was extended closer to 2100.

Since we used JSBACH\_FOM in stand-alone mode, air temperature and humidity are driver parameters. Thus, under the same climate scenario, these atmospheric variables remain unchanged between the different forest management scenarios despite the differences in their albedo and heat fluxes.

## 5 Conclusions

485

490

This study demonstrated that when EC, BA and PR were compared in terms of their total climate impact during 2010–2054 under, the lowest harvest intensity used in EC resulted in the most beneficial in terms of climate change mitigation. In turn, the highest harvest intensity resulted in PR having the weakest climate change mitigation benefit. The findings were similar in all three climate models CanESM2, MIROC5 and CNRM\_CM5 and under both RCP 4.5 and RCP 8.5.

Carbon sequestration was highlighted as the most prominent individual climate impact out of the climate impacts assessed in this study. Even though the less frequent harvests resulted in higher absorption of solar radiation, in terms of carbon equivalence, the albedo-induced impact was only in the range of 4–8% in comparison to the differences in carbon sequestration. The albedo-induced differences between the harvest scenarios were also counterweighed by the differences in the latent heat flux, which were of opposite sign and in the range of approximately 3–5% after accounting the sensible heat flux as well.

Code and data availability. The relevant model output data used in the study, and the code used to convert radiative forcing to carbon equivalent values, can be accessed from the Finnish Meteorological Institute's B2SHARE at: https://doi.org/10.57707/fmi-b2share.183309f91f4a48ceac70f05712714f97

Supplement. The supplement related to this article is available online at: https://doi.org/10.5194/bg-0-1-2025-supplement

# Appendix A

**Figure A1.** Change in the annual mean of woody carbon pool from 2010 to 2054 in CanESM2 in ecological (EC, blue), balanced (BA, black) and profit oriented (PR, orange) harvest scenarios under RCP 4.5 (solid lines) and RCP 8.5 (dashed lines) climate scenarios.

**Figure A2.** Change in the annual mean of AWEN (right) carbon pool from 2010 to 2054 in CanESM2 in ecological (EC, blue), balanced (BA, black) and profit oriented (PR, orange) harvest scenarios under RCP 4.5 (solid lines) and RCP 8.5 (dashed lines) climate scenarios.

## Appendix B

**Figure B1.** The incoming radiation profile used to determine the total albedo based on the monthly mean albedo values in the visible (orange) and near infrared (blue) ranges, and to showcase its influence on the differences in the amount of absorbed radiation between the harvest scenarios during different times of year.

# Appendix C

Figure C1. Monthly mean VIS albedo ten years before (solid) and 10 years after (dashed) a clear cut took place in the additional experiment.

Author contributions. JLe set up the manuscript with contributions from all authors and designed the visualization; LB designed the simulations and data processing and contributed to the visualization; JLe contributed to the data processing; JLi designed the harvest scenarios; JLe, LB and TM contributed to the analysis of the results and to the conceptualisation; all authors contributed to the writing process, TA acquired the funding and conceptualised the study.

Competing interests. The authors declare that they have no conflict of interest.

Acknowledgements. This work has been funded by Research Council of Finland project ForClimate (347794). We thank Research Council of Finland projects OPTICA (295874), C-NEUT (347860) and RESPEAT (341752). We also acknowledge the support from the Research Council of Finland through the Flagship program for Atmospheric and Climate Competence Center (ACCC, 337552).

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
