# Peer review of "Effect of future forest management on carbon and energy budgets in pine forests on mineral soil in southern Finland"

_EGUsphere, 2025_

## Referee Comment (RC1)

To the Editors and Authors,

This paper evaluates the climate impact of carbon sequestration, surface albedo, and turbulent energy exchange for three forest management scenarios of pine forests in Finland using a climate model. The topic is highly relevant and properly examined. Unfortunately, the manuscript suffers from poor language with low precision, and the current manuscript appears as a draft missing the clearness and conciseness needed at a final stage before publication. Sadly, this is reducing the dissemination of the content of the study. I recommend that the current version of the manuscript is rejected. However, I strongly recommend that the authors improve and revise major parts of the current manuscript before the new version should be submitted.

The Abstract suffers from a poor language lacking fluency and precision. I recommend that the authors improve the entire abstract. For example, some sentences expressed almost exactly the same, and there were lack of definitions (e.g., for JSBACH_FOM). The Abstract was also quite long. I recommend that the authors make the Abstract shorter and rewrite it to ensure that the content and the findings of the study are presented in a clear way.

Overall, I think the Introduction needs major revision. Generally, the language lacks fluency and precision, and parts of the content are not particularly relevant for the current study. I recommend that the authors work to improve the text and form the content in a way that aligns with the topic of the study (i.e., different climate impacts of forest managements). Particularly, since the authors' study compares climate impact from carbon against biophysical climate forcings, I think they should address the role of carbon in relation to forest management. For the subparagraph about albedo, they must improve connection between albedo and forest management. For example, in this subparagraph they elaborate a lot about "other land cover types" and albedo. I don't find this relevant at all or important for the current topic. Therefore, I recommend removing these parts. I am also missing more information about the role of turbulent fluxes as climate agents in boreal forests and/or for forest management as they have not been properly and sufficiently addressed. For example, are the exchange mechanisms of sensible and latent heat cooling or warming the climate? Why and how? How are these fluxes related to forest management? Moreover, the introduction does not address the concept of climate modelling for forests/forest management at all. This should be properly presented, e.g., by elaborating on forest-climate modelling of energy budgets and forest scenarios.

Whereas the text and the presentation are better in Methods and Discussion (still some textual errors), the Results suffer from too many details, making it difficult to extract the main findings of the study. I recommend that many of the figures are moved to Appendix, and that the surrounding text related to these figures are removed or shortened. Since the topic of the study is climate impacts of forest management, i.e., effects of carbon sequestration, shortwave albedo, and latent and sensible heat, I strongly recommend focusing and highlighting the findings related only to these impacts in the Results.

As I believe major parts of the study must be revised to improve language fluency and precision, there are several specific comments – such as language errors, lack of definitions, incorrect use of punctuations, and unclear wording – that I have not commented on or included in the specific comments following henceforth.

**Specific comments**

L2-3: This sentence lacks fluency. Please improve. I suggest "Carbon sequestration plays a central role for climate change mitigation in forest management. However, other climate impacts – such as turbulent fluxes and surface albedo – can significantly alter the climate mitigation of forests".

L5: What is JSBACH_FOM? This is not clear to me. Please explain/define.

L7-8: Please rewrite these sentences.

L29: Please define the term albedo.

L38-49: There are plenty of relevant studies, e.g.:
- Bright, R. M., et al. (2014). "Climate change implications of shifting forest management strategy in a boreal forest ecosystem of Norway." Glob Chang Biol **20**(2): 607-621.
- Kellomäki, S., et al. (2021). "Effects of different management options of Norway spruce on radiative forcing through changes in carbon stocks and albedo." Forestry: An International Journal of Forest Research **94**(4): 588-597.
- Kellomäki, S., et al. (2023). "Effects of thinning intensity and rotation length on albedo-and carbon stock-based radiative forcing in boreal Norway spruce stands." Forestry: cpac058.
- Bright, R. M., et al. (2024). "Relevance of surface albedo to forestry policy in high latitude and altitude regions may be overvalued." Environmental Research Letters **19**(9): 094023.
- Ramtvedt, E. N., et al. (2026). "Greater increase in surface albedo following clear-cutting than wildfire in pine dominated northern Swedish boreal forests." Agricultural and Forest Meteorology **376**: 110924.

L39: I believe the same albedo values apply to pine, spruce, and birch forests in Sweden and Norway as well. Consider rephrasing this, so it is shown that this also governs the general forests in Fennoscandia. Are the values reported for summer or winter? Please specify.

L38+40: I think these two sentences express almost the same. Please rewrite for better fluency and to avoid repetition.

L41: "tree species" should be change to "tree species composition" for better precision of language.

L43-44: Please review this sentence. How does the snow "accumulates" unevenly? Do you mean vertically or horizontally? Please specify. I don't think "perfectly" is the correct adverb here. Please improve the precision and be careful of what you mean.

L45-47: Are you sure that only the effect of decreasing snow cover is causing this? This is a bit vague. I think this is more complex. What about seasonal snow cover duration for example?

L47: Can this ("which will potentially reduce the differences in winter albedo between forests and other land cover types") be documented by a reference? I find this a bit speculative. Moreover, do you know the without-snow winter albedo of these other land cover types?

L58: What species of Finnish pine forests. Please be precise and add species name.

L73: What is MPI ESM?

L90-97: I don't understand this. What is your definition of albedo? I thought it was shortwave solar radiation? But you don't define it. Spectral wavelengths are missing. Why is VIS and NIR albedo calculated? I think the heading for this subsection is misleading because the subsection does not address radiation balance at all, only albedo.

L133: Why do you include data from all these climate models? This is never motivated, so the readers are left wondering why.

Fig. 2, 3, and 4: Consider changing the numbers on the x-axis to e.g., every 20 year (instead of every 5 year as currently presented) for better readability. Also consider to place these figures next to each other for easier comparison.

L272: Figure A1 and A2 should not only be mentioned, but it should properly be described what the trends in woody carbon pool ++ show.

Fig. 6: I don't think changes in LAI are that interesting or relevant for the aim of this study. I recommend moving this figure and related text to the Appendix.

L280: Section 3.2.1: I am not sure it is a good idea to present the albedo separately for VIS and NIR. In the context of climate impact, we are interested in the broadband integrated albedo (i.e., SW = NIR+VIS). I think the current presentation with NIR and VIS albedo can be confusing for the broader group of readers (without expertise in radiation), at least when the terms are not defined and clearly presented. I recommend changing this part.